

# Thermal conditions and age structure determine the spawning regularities and condition of Baltic herring (*Clupea harengus membras*) in the NE of the Baltic Sea

Timo Arula, Heli Shpilev, Tiit Raid and Elor Sepp

Estonian Marine Institute, University of Tartu, Tallinn, Estonia

## ABSTRACT

Baltic herring (*Clupea harengus membras*) is a total spawner with a group-synchronous ovarian organization. Age polymodality in total spawners is considered an important factor in assuring that a strong population is sustainable under an intensive harvesting regime and different climatic conditions. In the present study, we investigated the seasonal and inter-annual variation in spawner age structure and the effect of preceding winter thermal conditions on the start of the herring spawning and larvae retention period. Herring spawning season in the Gulf of Riga starts up to six weeks later after colder winters compared to milder winters. Significantly older individuals dominated at the beginning of the spawning season, and thus herring mean age gradually decreased towards the end of the spawning season from 1999–2015. On an annual scale, this pattern was obvious after cold winters, while after mild winters the pattern did not continue, indicating a more homogenous maturation cycle and spawning period, despite the age and size of the herring population in mild winters. Further, herring condition factor was studied in relation to age and spawning season following different winter thermal conditions. Young, 2- and 3- year old first-spawning herring experienced significantly lower conditions after cold winters compared to older ages, indicating an age-dependent effect of preceding winter on herring maturation cycle, condition and spawning time.

# INTRODUCTION

Age structure of spawning populations, spawning seasonality, and migration to spawning grounds have been widely studied in relation to fishing regulations for population conservation and climate variability (*Lambert, 1987*; *Sims et al., 2004*; *Sissener & Bjørndal, 2005*; *Margonski et al., 2010*). A fish population that is sustainable in the long run contains a wide diversity of age classes within the limits of species-specific life span (*Planque et al., 2010*; *Brunel & Piet, 2013*). For Baltic spring spawning herring (*Clupea harengus membras*), there have been no studies focusing on the age structure of spawning population, although such material has been collected with high precision for decades for routine stock assessment

Corresponding author
Timo Arula, Timo.Arula@ut.ee

(*ICES, 2018*). Within the same spawning season, many different spawning groups of herring exploit same spawning ground, migrating to spawning grounds in "spawning waves". Predictions of such spawning waves provide resource managers grounds to plan commercial fishing to conserve certain age classes (*Sissener & Bjørndal, 2005*). Age-specific spawning waves would appear to be the result of differential maturation with older, returning individuals spawning first, and younger, first time spawning groups progressively later (*Rannak, 1971*; *Ojaveer, Raid & Suursaar, 2004*; *Raid et al., 2010*). Still, fish spawning regularities vary between years and are hardly predictable based exclusively on age structure (*Sims et al., 2004*; *Sissener & Bjørndal, 2005*). Fish tend to have certain adaptation plasticity in the timing of life cycle events such as reproduction and in the length of spawning seasons as a response to environmental cues (e.g., photoperiod and temperature, *Geffen, 2009*).

Spawning timing is a result of differences in maturation as a result of age- and size-group - specific energy allocations which determine individual condition (*Lambert & Messieh, 1989*; *Slotte, Johannessen & Kjesbu, 2000*). However, such patterns may be modified by annually varying hydro-climatic conditions interacting with feeding conditions in overwintering areas, a mechanism that remains poorly studied (*Kjesbu, 1994*). Thus, the spawning regularities of age classes dominating in a mature component of the fish stock occur in "spawning waves" (*Lambert, 1990*). Such a spawning pattern is believed to be an effective tactic to secure higher survival probability for the first-feeding larval stage because they are distributing their hatching over a longer period of time (*Secor, 2000*). To analyze and understand this pattern accepts complex studies that include detailed observation data of spawning populations and the surrounding environment—material that may not exist for some commercially harvested fish populations (*Secor, 2015*). Based on long-term observations it has been demonstrated that in years when the spawning season was of longer duration, more abundant year-classes of herring were produced (*Dragesund, 1970*; *Ojaveer et al., 2011*). The habit of herring spawning over a longer period leading to a succession of larval cohorts is a bet-hedging tactic which enhances the probability to match larvae in space and time with a highly variable prey field (*Lambert & Ware, 1984*). Thus, a higher number of spawning waves/larval cohorts over time should secure a more abundant year-class.

Baltic herring contributes, an average of 42% of commercial fishery in the Baltic Sea (*Raid et al., 2015*). Total landings of herring in the Baltic Sea have stabilized during recent years to around 200–250 Kt after being under 200 Kt in 2004–2005 (*Raid et al., 2015*). Baltic herring spawn in coastal areas, where high food and oxygen concentrations create a good environment for successful hatching and ontogeny of larvae (*Raid, 1985*; *Arula et al., 2012*; *Arula et al., 2015*; *Arula et al., 2016*). After spawning in late spring and early summer, adults migrate to deeper waters to feed and improve their fitness before winter (*Ojaveer, Pihu & Saat, 2003*). The coastal areas (especially Pärnu Bay) in the NE of the Gulf of Riga (GoR) serve as a suitable spawning environment for the GoR herring population, with the adjacent coastal areas being the most intensely utilized spawning and nursery grounds (*Ojaveer et al., 2011*).

The GoR herring spawns from mid-May to mid-July at temperatures from 9 to 16 degrees C, mainly on demersal vegetation, gravel, etc. The depth of the main spawning

grounds varies from 2 to 5 m. Spawning of the GoR herring is temporally separated from its neighboring population—the open-sea herring of the NE Baltic, belonging to the Central Baltic herring stock. Central Baltic herring population spawns earlier around the Saaremaa and Hiiumaa islands (from the middle of April to the middle of June) and at a lower temperature (5–13 degrees C) than the GoR herring. However, some overlapping occurs because part of the Central Baltic herring population spawns in the eastern areas of the gulf, including Pärnu Bay (*Ojaveer, 1988*).

Winter (January–March) air temperatures in the NE part of the GoR have increased from an average of −14.6 degrees C in the late 1940s to −8.6 degrees C in the late 1980s (*Ojaveer et al., 2011*; *Arula et al., 2014a*). Cold winters, dominated by easterly winds, are unfavorable for Baltic herring feeding and gonadal maturation. Cold temperatures affect the accumulation of energy before herring reproduction because herring do not feed at temperatures below 2 degrees C (*Ojaveer et al., 2011* and references therein). The maturation and timing of copepod production peak, the major food source of herring larvae, is also hindered by winter temperatures (*Arula, Ojaveer & Klais, 2014b*). These circumstances motivated us to compare data collected in the GoR spring herring spawning season from April to July and larval herring retention from May to August in 1947–1986 and 1999–2015 and to specifically ask: (i) has the start of spawning season shifted significantly earlier in parallel with warming since the late 1980s; (ii) is the start of spawning season determined by the preceding winter air temperatures and population age structure; (iii) does variation in thermal conditions and spawning time result in different individual conditions between first- and repeat spawners?

## MATERIALS & METHODS

### Structure of spawning schools

Individual herrings considered in analyses were regularly collected from commercial trap-net landings stationed on herring spawning grounds or in adjacent areas during the fishing period from April to June (corresponding to calendar weeks 15–25) from 1999–2015 (Table 1). Total length, total weight, age and maturity stage were determined from 12,148 individuals in total (including recruitment, age 1 abundance; Apendix 1). Gender and maturation stage were determined using methods developed by the Workshop on Sexual Maturity Staging of Herring and Sprat Working Group (*ICES, 2011*). The scale includes six maturity stages: (I) immature; (II) maturing; (III) spawning; (IV) spent; (V) regeneration; and (VI) abnormal. The samples analyzed in the present study contained individuals in the pre-spawning and spawning stages (hereafter: spawning) that are ready to spawn.

The start of herring spawning season was calculated based on data collected in herring trap-net landings with a weekly accuracy. The start of herring spawning season (hereafter: start of spawning) was determined by the appearance of individuals in the spawning stage found in spring commercial catches. Because Baltic herring spawning season starts shortly after the ice break in Pärnu Bay and herring do not spawn under ice, any spawning earlier than observed in the present study could not occur. National regulations determine the annual allowable catch (quotas) for spawning herring trapnet landings, and therefore

Arula et al. (2019), *PeerJ*, DOI 10.7717/peerj.7345

**Table 1** **Number of matured individuals ( $N = 12031$) collected from trap-net catches and analyzed in present study.**

| Year | 99 | 00 | 01 | 02 | 03 | 04 | 05 | 06 | 07 | 08 | 09 | 10 | 11 | 12 | 13 | 14 | 15 |
|------|-----|-----|-----|-----|-----|-----|-----|-----|-----|-----|-----|-----|-----|-----|-----|-----|-----|
| *Number of individuals analysed* | 794 | 777 | 642 | 706 | 587 | 533 | 439 | 585 | 654 | 617 | 669 | 998 | 647 | 862 | 994 | 741 | 786 |

competition between herring fishermen exists during trapnet catch season. Herring fishermen landings are restricted with total allowable quotas given each year, based on herring spawning stock biomass. Therefore, fishermen targeting herring spawning schools start shortly after the ice break and always before abundant herring spawning schools migrate to Pärnu Bay. Trapnets are regularly inspected by fishermen in that period and, due to regular fisheries monitoring obligatory fish research, communication regarding presence/absence of spawning herring in trapnets is available for researchers.

In order to compare the start of spawning season in Pärnu Bay before and after 1990, the number of years that were available in each decade were used: 1947–49, 1953–1955, 1964–66, 1974–76, and 1984–86 (for details see *Ojaveer et al., 2007*). Since the late 1980s, the winter air temperature increased substantially compared to former periods and this has markedly changed the spawning stock biomass, recruitment and larval herring dynamics in the GoR (*Ojaveer et al., 2011*; *Arula et al., 2014a*). As we aimed to analyse whether the start of spawning season was significantly different before and after the climate driven regime shift in the late 1980s, and because a limited selection of years prior to the 1990s are covered by data (which may not represent "the average" conditions for each decade), "winter air temperature" and "YEAR" were included as covariates. Therefore, prior to comparing the start of spawning season before and after the 1990s, the homogeneity of slopes for categorical variables for both periods were controlled and compared to achieve adjusted mean cw for the start of the spawning season. Analysis of covariance (ANCOVA) was applied to test the main and interactive effects of categorical variables (winter air temperature and YEAR) on a continuous dependent variable (start of spawning, cw), controlling for the effects of selected continuous variables, which co-vary with the dependent.

## Herring larvae

Herring larvae were obtained at nine stations with a depth ranging from 3 to 10 m from early May to late August, depending on larval occurrence, covering the same area where spawning herring samples were collected. Hauls were performed with the Hensen larval fish net (mouth opening $d = 80$ cm, mesh size $= 500$ m, and codend $= 300$ m) from the near-surface layer (0–1 m) horizontally by 10-minute hauls at a speed of approx. 2 knots and the volume of water filtered during each haul (measured with flowmeter Hydro-Bios "Digital Flowmeter 483110").

Data for the first appearance and the end of the larval herring occurrences were calculated according to methodology described in (*Arula et al., 2014a*) (Apendix 2). These were calculated from the cumulative sums of weekly abundances, with the points reaching 10% and 90% from the annual sum, respectively, from 1999–2015 (for details see *Arula et al., 2014a*). The length of the larval herring occurrence period was defined as the number of weeks from the first appearance to the end of the season (for details see *Arula et al., 2014a*). The end of herring spawning season was determined as the previous calendar week of the last abundant presence of recently hatched yolk-sac (Sl 6–7 mm) herring larvae. Since water temperatures had increased above l7 degrees C, embryonal development (fertilization of eggs to hatching the yolk-sac larvae) took approximately one week (*Ojaveer, 1981*).

## Environmental data

Water temperatures of herring spawning grounds have been quoted as the main characteristic determining successful development of viable embryos and survival of larvae at hatch (*Ojaveer, 1981*). Data from weekly measurements during larval fish surveys were obtained to characterize suitable temperature conditions for embryonic development and survival. The percentage of hatched normal larvae of spring spawning herring in the GoR depended on water temperature. According to *Ojaveer (1981)*, the optimal water temperature of spawning grounds that assure normal embryonal development is between 7 degrees C and l7 degrees C for the Gulf of Riga herring. In temperatures above l7 degrees C, embryonal mortality of spring spawning herring was 35% and increased at higher temperatures.

The sum of monthly mean air temperature (hereafter –winter air temperature) at Pärnu weather station (Fig. 1) in the period of January–March was used as background information. In the shallow Gulf of Riga, the effect of winter severity on thermal regime of the vertical water column is very variable. In the severe winters (presented in our study as 75% percentiles) the whole sea surface is covered by ice, while in mild winters (presented in our study as 25% percentiles) the ice cover occur only on shallow bays with the rest of the sea remaining ice-free, allowing to conclude that favorable water column for herring maturation and their prey overwintering is remarkably broader in mild winters and narrow during severe winters (*Jaagus, 2006*). Winter air temperature was used as a proxy value to estimate the potential shift in spawning timing due to starvation due to overwintering, and the effect on individual conditions after severe winters. This has been shown to cause considerable overwintering losses and size-selective mortality among young-of-the-year small pelagic species as well as larger predators, like striped bass in northern seas (*Martino & Houde, 2012*; *Anderson & Scharf, 2013*). Many studies demonstrate that decreased food availability in winter cause fish to starve and exhaust their energy reserves (*Hurst, 2007* and references therein). At the same time, at extremely cold temperatures, acute thermal stress can disrupt osmoregulatory function and become the primary cause of death (*McCollum, Bunnell & Stein, 2003*).

To test the effect of winter air temperature on the start of spawning season and changes in age structure, a polynomial regression model was implemented in software PAST (*Hammer, Harper & Ryan, 2001*), considering the best-observed pattern of the model fit. First, the linear model was applied, and if a curve model performed significantly better than a simple linear model by taking into account the additional parameters used for smoothing, curve fit regression (polynomial fit) was applied. The maximum number of degrees of freedom was restricted to 3 ($k = 4$) to avoid overfitting. The lowest Akaike information criterion (AIC) was obtained with one common trend. The assumptions for normality (Shapiro–Wilk or Kolmogorov–Smirnov), constant variance, and independent residuals were tested before applying polynomial regression. Regression models included: the start of herring spawning season and mean age of spawners (dependent variables), and winter air temperature and calendar week (independent variables).

A non-parametric Mann–Kendall trend test was used to detect the presence of long-term trends in annual-scale variables: (a) spawning time and (b) larval onset. The non-parametric

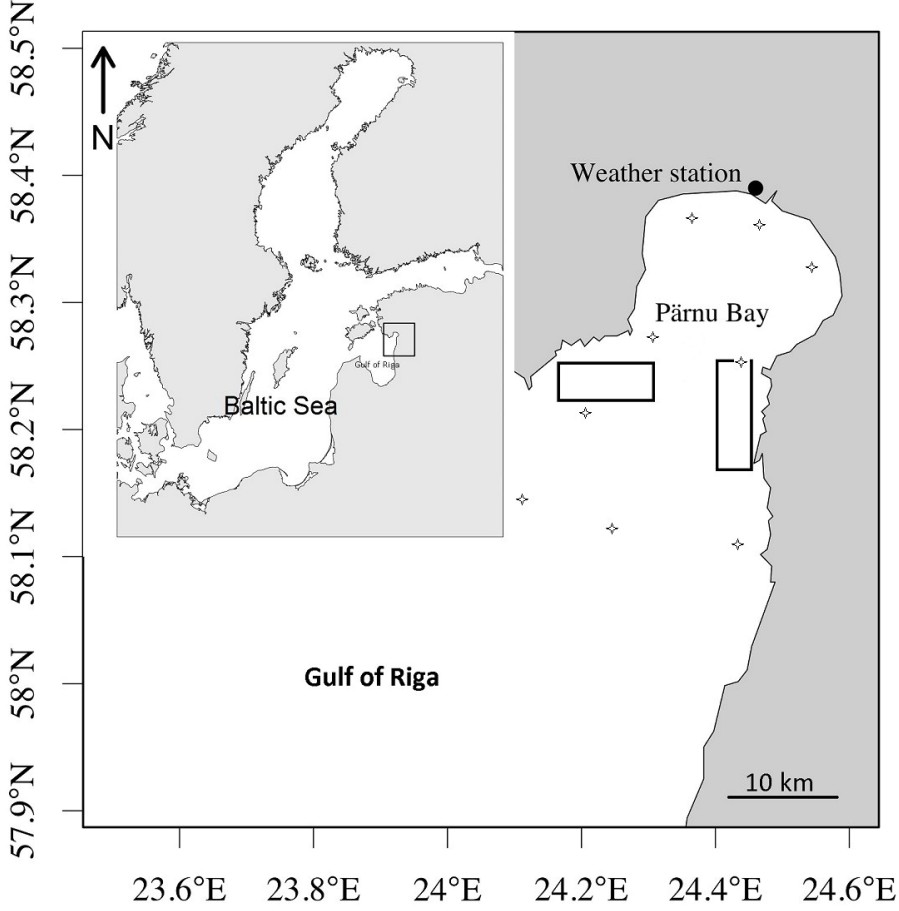

**Figure 1** **Rectangles showing the area of herring commercial catches and data sampling.** Rectangles showing the area of herring commercial catches and data sampling to characterize herring spawning. Stars denote stations of larval herring collection and water temperature measurements.

Mann–Kendall test was chosen because of the non-linear appearance of most of the trends. Before application of trend test, autocorrelation (AC) was controlled in the response variables.

## Data aggregation

Age and condition of fish may affect spawning timing, therefore, we hypothesized that older herrings, having a higher condition factor (K) value, spawn earlier. We hypothesized that despite age, herring K in spawning season is affected by the previous winter air temperature and the effect differs between young and old herrings. We analyzed herring K by age: young herring that are first-time spawners (ages 2–3) were compared to old herring that were repeated spawners (ages 4-10+) to determine if there is a significant difference amongst and between young- and old herring K after the warmest and the coldest winters.

Finally, we tested whether K affects the herring spawning season. To do this, we split the spawning season into early and late periods (early (prior cw 20) and late (after cw 21)), and compared herring K accordingly.

To study the effect of preceding winter air temperature, we aggregated data accordingly: coldest and warmest winters. For such comparison, we used 25% and 75% percentiles of winter air temperatures from the years 1999-2015 and selected the years accordingly.

## Fulton condition

Fulton's condition factor (K, Eq. (1)), which assumes that the total weight of a fish is proportional to the cube of its length, was used to measure each individual fish's health (*Nash, Valencia & Geffen, 2006*):

$$K = 100 * TW/TL^3 \tag{1}$$

where TW is total body wet weight in grams and TL is total length in cm; the factor 100 is used to bring K close to a value of one.

To test any differences in median values of K between young and old individuals and/or after mild and cold winters, one-way analyses of variance (ANOVA) were used. First, data were analyzed for normality of distribution (Wilkinson–Shapiro tests) and equal variance. If the assumptions for Equal Variance were met ($P > 0.05$), a Kruskal–Wallis One Way Analysis of Variance on Ranks Test was run. If the equal variance test failed ($P < 0.05$) a one way ANOVA on ranks Tukey test was run, which is more conservative than the Student–Neuman–Keuls Test, i.e., it is less likely to infer that a given difference is statistically different. If the assumptions for normal distribution were not met, Dunn's method in the Kruskal-Wallis one way analysis of variance on ranks test was applied for pairwise comparisons using SigmaStat 12 software (Systat Software, San Jose, CA, USA). For the statistical tests, alpha was set at $< 0.05$.

The numerical values of the spawning stock biomass of the GoR herring population were taken from publicly accessible stock assessment reports based on catch statistics and regular biological analyses for the determination of age, length, weight, maturity and other relevant parameters (*ICES, 2018*; Apendix 3).

## RESULTS

### Spawning season

The start of spawning season showed interannual variability, occurring between calendar week 15 and 20, corresponding with early April and mid-May, respectively (Fig. 2). There was no temporal trend in the start of spawning seasons from 1999–2015 (MannKendall's tau $= 0.89$, $p = 0.37$). Spawning season lasted from six to twelve weeks and its duration was insignificantly affected by the start of spawning season. The first appearance of herring larvae varied between calendar week 18 and 22, corresponding to early May and early June, respectively. Similar to the start of spawning season, there was no temporal trend in larval herring first appearance in 1999–2015 (Mann-Kendall's tau $= 0.48$, $p = 0.63$). Earlier larval first appearance corresponded to an earlier start to spawning season (polynomial regression; $N = 17$, AIC $= 14.60$, $F = 6.886$, *adj.* $R^2 = 0.68$, $P < 0.001$). In the years of earlier spawning, the time lag between the start of spawning season and larval first appearance was significantly longer, attributable to the longer duration of embryonal

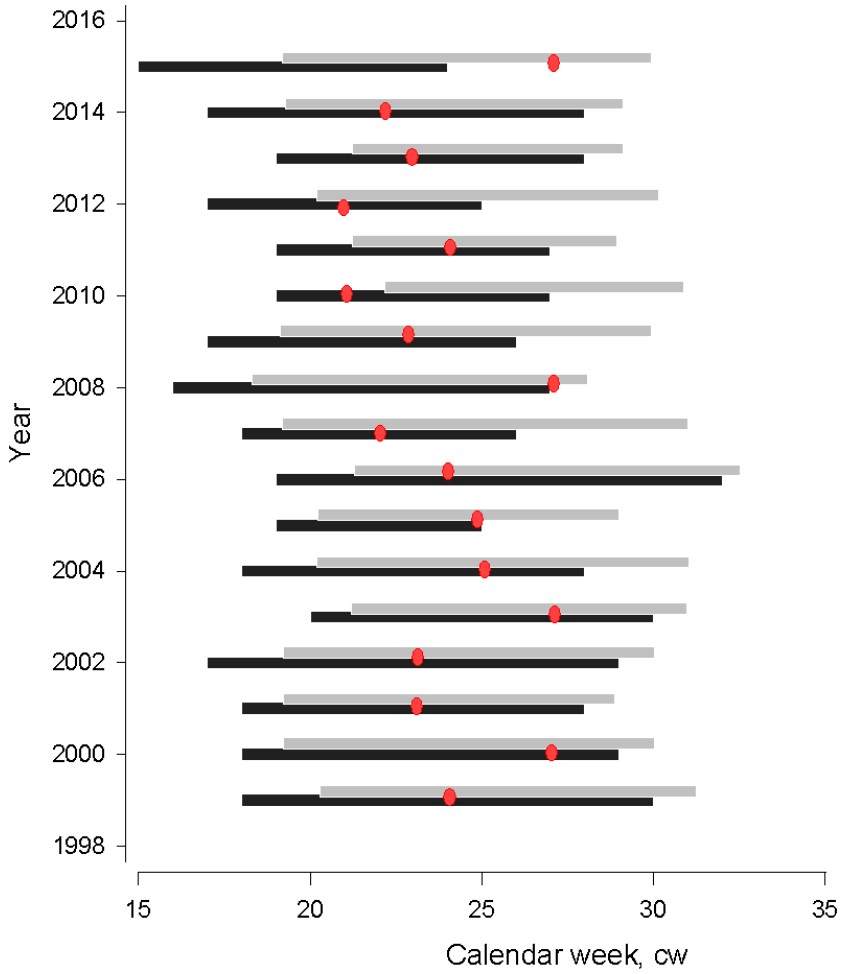

**Figure 2** **Gulf of Riga herring spawning (black bar) and larvae (grey bar) distribution seasons.** Gulf of Riga herring spawning (black bar) and larvae (grey bar) distribution seasons in 1999–2015. Red dots denote the time when water temperatures exceed 17 °C, which is considered critical for normal embryonic development (*Ojaveer, 1981*).

development until hatching as a response to colder temperatures (polynomial regression; $N = 17$, AIC $= 12.57$, $F = 4.66$, *adj.* $R^2 = 0.40$, $P < 0.05$).

The time period when water temperatures of spawning grounds increased to over 17 °C and resulted in high levels of embryonic mortality varied from cw 21 to 27. In most years, temperatures became critical for embryonal development in the middle of the spawning season, with an exception in 2015 when 17 degrees C temperatures were observed several weeks after herring spawning.

## The effect of winter air temperature

We did not observe a significant difference in the start of spawning season prior to and after the late 1980s when using winter air temperature and YEAR as covariates (ANCOVA; $F_{1;29} = 2.101$, $P > 0.05$). While winter air temperature contributed significantly in the

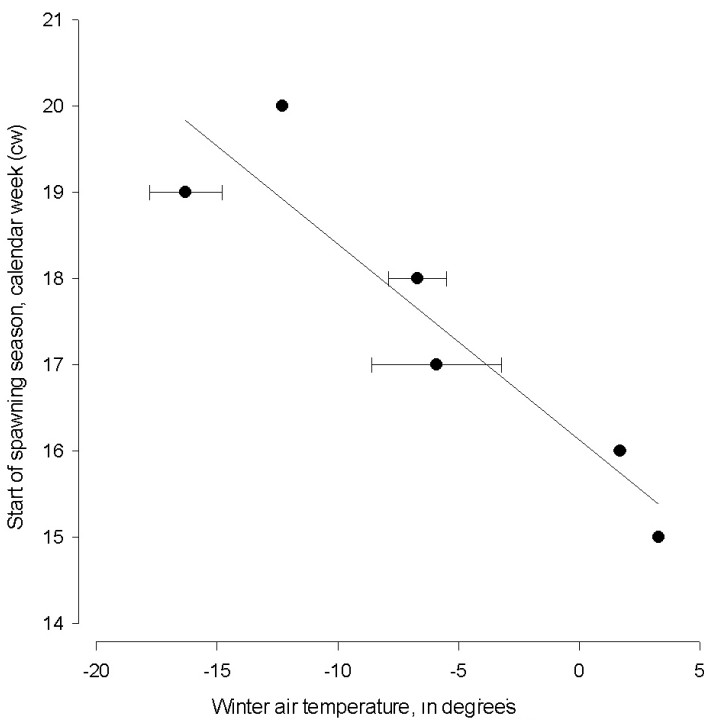

**Figure 3** **The linear regression between the start of herring spawning season and sum of mean monthly winter air temperature.** The linear regression between the start of herring spawning season and sum of mean monthly winter air temperature in 1999–2015. Error bars = 1 s.e.

model (ANCOVA; $F_{1;29} = 7.106$, $P < 0.05$), the YEAR effect remained insignificant (ANCOVA; $F_{1;29} = 1.450$, $P > 0.05$).

Winter air temperature affected the start of spawning season from 1999–2015 (linear regression; start of spawning = $16.128 - (0.227 *$ winter air temperature), $N = 6$, AIC = 44.13, $F = 25.123$, *adj.* $R^2 = 0.83$, $P < 0.001$, Fig. 3). The spawning season following cold winters began in mid-May (cw 20), which is six weeks later than the start of spawning season following the warmest winter.

Herring spawning stock biomass and the number of old herrings (abundance of age group 4+) varied markedly in an annual scale. The lowest spawning stock biomass was observed in 2006 and the highest in 2014 (69.9 and $119.6 \times 10^3$ tonnes, respectively) from 1999–2015 (Fig. 4). Number of old herrings varied between 13.0 and $27.3 \times 10^6$ individuals. Neither GoR herring spawning stock biomass nor the number of old herrings from 1999–2015 significantly explained the start or duration of herring spawning season (polynomial regression; $N = 17$, $p > 0.05$).

### Herring age structure during the spawning season

The changes in the mean age of herring displayed a seasonal pattern with the oldest, repeat spawners dominating at the start of spawning season, and young, first time spawning individuals prevailing towards the end of the season (polynomial regression; mean age

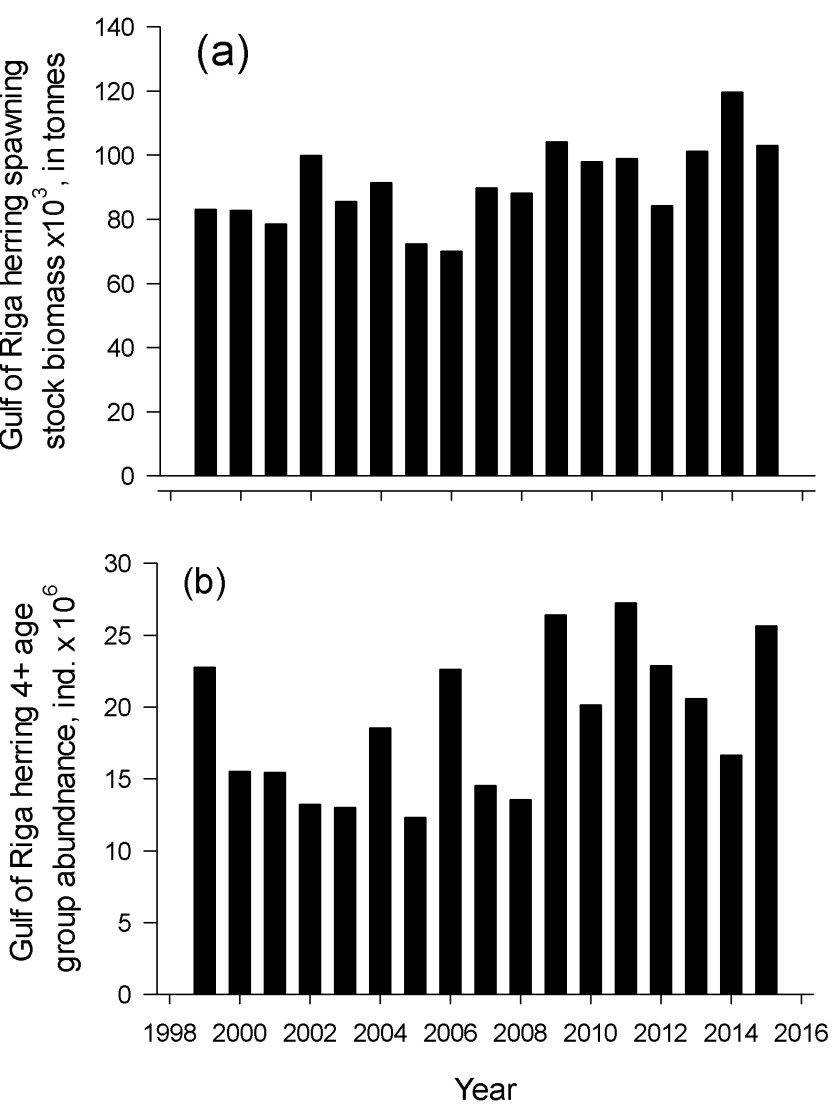

**Figure 4 Dynamics of spawner biomass and abundance of age 4-10+.** The temporal dynamics of the Gulf of Riga herring spawning stock biomass, Spawning stock biomass (A) and age 4-10+ individuals abundance (B) in spawning stock in 1999–2015 (*ICES, 2018*).

$= 2.402 + (0.214\ {}^\star\text{calendar week}) - (0.00698\ {}^\star\text{calendar week}^2)$ $N = 11$, AIC $= 6.02$, $F = 146.89$, *adj.* $R^2 = 0.95$, $p < 0.001$, Fig. 5).

The correlation between the progression of herring average age and calendar week during spawning period by each year from 1999–2015 varied in a broad scale (Pearson correlation coefficient: $R = -0.04$ to $-0.84$). The strength of correlation between the average age of spawner and cw was determined by preceding winter air temperature (polynomial regression; $N = 16$, AIC $= 394.55$, $F = 6.86$, *adj.* $R^2 = 0.36$, $P < 0.05$). The strength of correlation improved remarkably in the years with the coldest winters and it was relatively moderate in years with warm winters, indicating the importance of preceding environmental conditions in shaping changes in herring age structure during spawning.
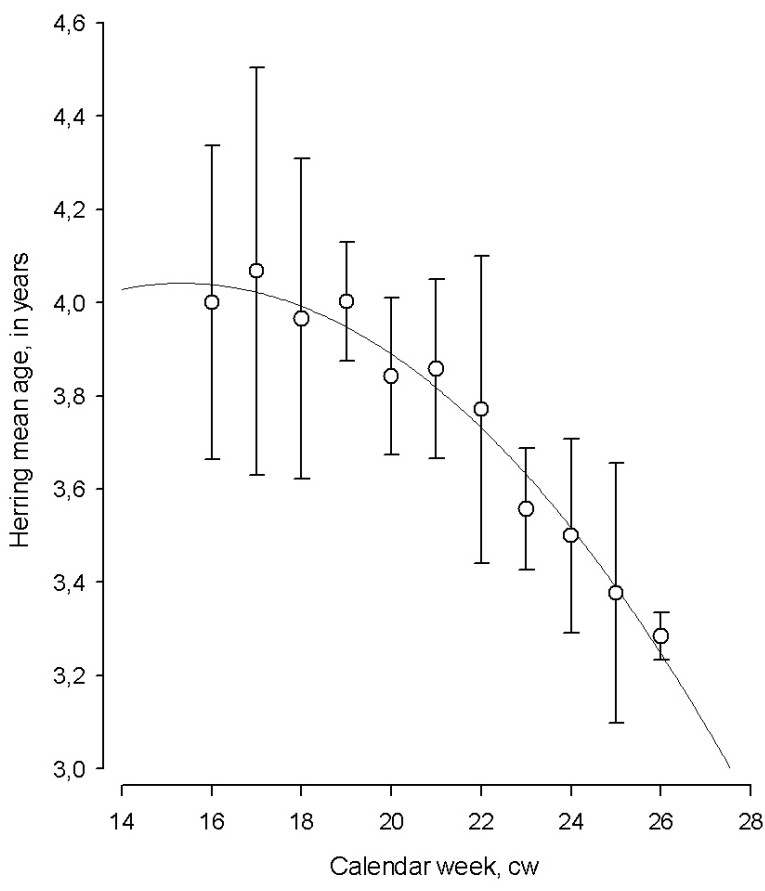

**Figure 5   The temporal dynamics of Baltic spring spawning herring mean age during the spawning season.** The temporal dynamics of Baltic spring spawning herring mean age during the spawning season. Each data point in particular calendar week (cw) denote average age for 1999–2015. Error bars = 1 s.e.

## Variation in herring individual condition factor (K)

The largest difference in condition factor occurred between young herring after cold winters and old herring after warm winters (Kruskal–Walls one-way ANOVA; $F_{(1;46)} = 9.98$, $P < 0.01$, Fig. 6). Also, young herring were in better condition after warm winters than after cold winters (Kruskal–Wallis one-way ANOVA; $F_{(1;46)} = 5.69$, $P < 0.05$), while old herring had similar K despite winter air temperature (Kruskal–Wallis one-way ANOVA; $F_{(1;48)} = 1.38$, $p > 0.05$).

Despite the timing of the spawning season, within age groups the condition was similar (Kruskal–Wallis one-way ANOVA; $P > 0.05$; Fig. 7). Only old herring spawning in late in the season had a higher condition, compared to early spawning young herring (Tukey Test one-way ANOVA; $Q = 3.43$, $P < 0.01$).

## DISCUSSION

In the present study, we aimed to investigate GoR spring herring spawning seasonality and changes in age composition throughout the spawning season from 1999–2015, as well as

Arula et al. (2019), *PeerJ*, DOI 10.7717/peerj.7345    12/22

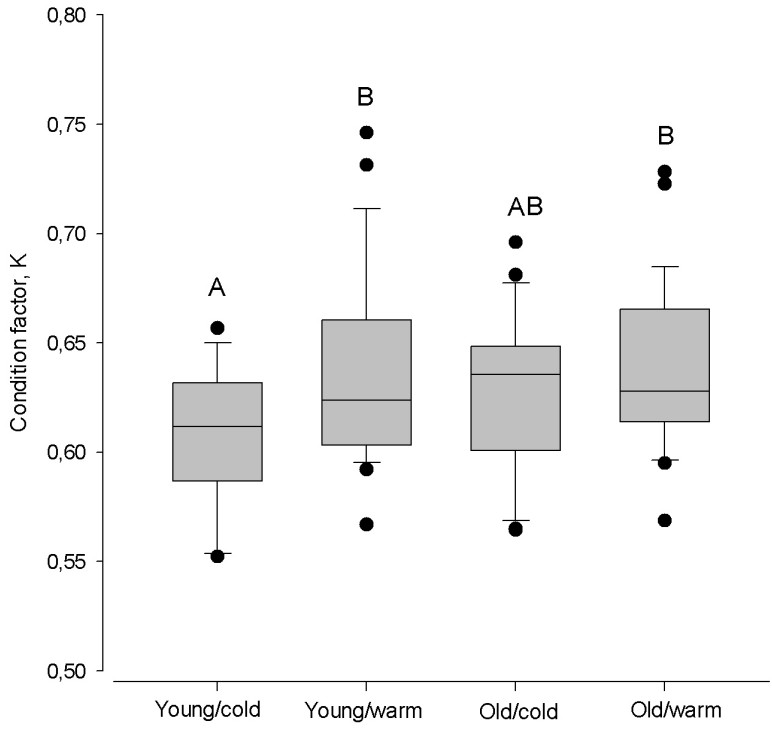

**Figure 6** **Box plots of the Fulton condition factor (K) of young (2–3 year) and old (4 − 10+ year) her-ring after cold and warm winter (25% and 75% percentiles of winter air temperatures over the years in 1999–2015).** Vertical boxes denote quartiles, the line inside the box median, whiskers 10th and 90th per-centiles, and dots show extreme values. Different letters indicate $p < 0.05$.

to analyze the effect of preceding winter air temperature and K on spawning seasonality. We found that preceding winter air temperature influenced herring spawning and larval first appearance. The spawning season began up to six weeks earlier after the mildest winter compared to the coldest winters. There were no significant differences in the start of spawning season before and after the climatic regime shift beginning in the late 1980s in the Baltic Sea. It seems that several observed warmer winters in recent period contribute to remarkably earlier herring spawning and in former period, despite of several very cold winters, herring spawning was not remarkably affected. In other words, warmer than average conditions in winter seem to have significant effect on herring earlier spawning, while colder than average winters do not seem to affect the start of herring spawning. The K amongst young or old individuals was similar despite spawning time, while the winter air temperature had a significant effect by age group. Young herring displayed poorer K compared to old herring after cold winters, indicating that first-time spawners are more sensitive and influenced by preceding winter conditions.

Winter air temperature preceding to herring spawning migration affected the start of spawning season by shifting it up to six weeks earlier after the mildest winters compared to the coldest winters. The mechanism governing such a pattern might be that of the shallow gulf, where herring and its prey's overwintering habitat is situated in the areas

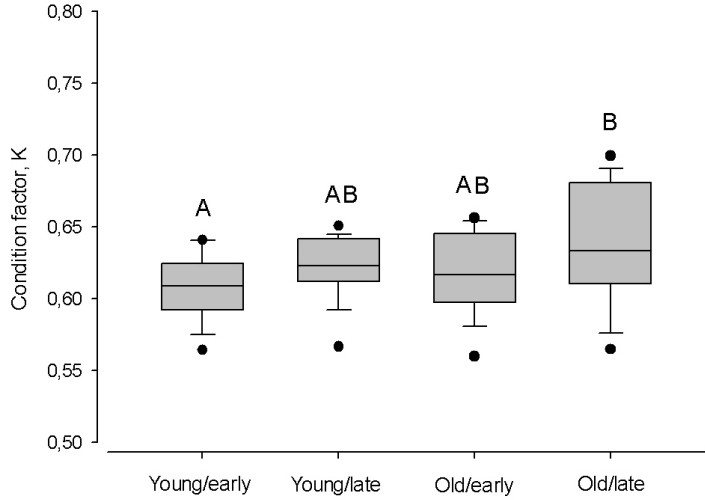

**Figure 7** **Box plots of the Fulton condition factor (K) of young (2–3 year) and old herring (4 − 10+ year) in early (prior cw 20) and late (after cw 21) spawning season.** Box plots of the Fulton condition factor (K) of young (2–3 year) and old (4 − 10+ year) herring after cold and warm winter (25% and 75% percentiles of winter air temperatures over the years in 1999–2015). Vertical boxes denote quartiles, the line inside the box median, whiskers 10th and 90th percentiles, and dots show extreme values. Different letters indicate $p < 0.05$.

where the longest temporal extent of ice cover occurs. Winter air temperature determines water thermal regimes in winter and controls the success of herring reproduction by mainly limiting habitat conditions of gulf herring in winter (*Ojaveer, 1988*). The area and distribution of the thermophilic organism are affected, especially of those that inhabit the NE region of the Baltic Sea. We chose winter air temperature because cold winters have been observed to limit the volume of water in shallow gulfs in which wintering, feeding and preparation for reproduction (maturation) of herring occurs (*Ojaveer & Kalejs, 2010*). In the Estonian coast winter severity is variable, and in severe winters (presented in our study as the 75% percentile) the whole sea surface is covered by ice, while in mild winters (presented in our study as the 25% percentile) the ice cover exists only on shallow bays, and the rest of the sea is ice-free (*Jaagus, 2006*). This allows us to conclude that the favorable water column conditions for overwintering herring and their prey is markedly wider in mild winters and limited in severe winters, reducing the probability for good overwintering conditions for faster maturation. Such a high variability allows one to assume that winter air temperature should be a sensitive indicator of extensively variable local climate conditions.

Changes in spawning location and timing could result in early life stage dispersal to inadequate habitats for the continuation of the life cycle (*Secor, 2015*). Alternatively, consistent spawning locations and timing could interact with changing thermal environments and result in larvae not arriving at suitable habitats for the continuation of the life cycle (*Shoji et al., 2011*). In the present study, we observed considerable inter-annual variation without a long-term pattern at the start of spawning season. The start of spawning season varied by six weeks and was strongly linked to the preceding winter air temperature,

with more severe winters resulting in delayed spawning and hatching of larvae. Delayed hatching of fish larvae result temporal mismatch with suitable prey, causing elevated mortality rates amongst fish recruitment cohort (*Houde, 1989*; *Arula et al., 2014a*; *Arula, Ojaveer & Klais, 2014b*; *Arula et al., 2016*). Such changes in prey and predator overlap might affect the survival of larvae (*Llopiz et al., 2014*), and a shift in spawning may result in a change in the productivity of spawning stock biomass. *Genner et al. (2010)* found a correlation between changes in the timing of larval occurrence of several spring spawning fish species and temperature in the English Channel: spawning occurred earlier with increasing temperatures for spring spawners. Despite significant effects of winter air temperature on herring spawning in the NE of the GoR, we did not observe a significant trend from 1999–2015 or significant difference in the start of spawning before and after the 1990s, indicating high plasticity of gulf herring spawning.

Changes in adult and early life stage behavior can affect connectivity, recruitment, and population biomass (*Secor, 2015*). Adults of many species make large migrations to specific spawning grounds during the same season year after year, presumably targeting the most favorable spawning and nursery grounds to increase the chances of larval survival (*Ciannelli, Bailey & Olsen, 2014*; *Secor, 2015*). Therefore, it is believed that population biomass variability could be a response of spawning behavior to ensure success of survival in early life stages (*Hjort, 1914*; *Houde, 2008*; *Peck et al., 2012*). The first appearance of fish larvae is determined by the timing of spawning, as shown in the present study. *Lambert (1987)* and *Lambert (1990)* found that spawning duration of herring is dependent on the number of spawning waves, which is dependent on the number of year classes present in spawning stock. We did not find either GoR herring spawning stock biomass or the abundance of old repeat spawners to correlate with the start of spawning nor the duration of spawning season. In contrast, for Icelandic summer-spawning herring, it was shown that spawning occurs on average seven days later when spawning stock biomass is greater than the long-term average, which might be the result of higher food competition and therefore slower ovary growth (*Óskarsson & Taggart, 2009*). It could be expected that if spawning season starts later, it also ends later than in conditions when earlier spawning occurs. However, there was no such consistent pattern in our study, indicating that Baltic herring spawning duration is mainly modified by the age structure of spawners—temperatures during the spawning season and a later start to spawning in the season will not have any effect on the duration or ending date. Also, regardless of the start of spawning season, herring embryos had similar chances of exposure to unfavorable conditions in terms of water temperature exceeding their thermal tolerance (*Ojaveer, 1981*).

Gonad maturation is thermally regulated and is proposed to drive the spawning and larval appearance of marine fish (*Lange & Greve, 1997*). Several studies have demonstrated how gonadal maturation rates improve in warmer environments. For example, in Lake Geneva, the precise time of roach (*Rutilus rutilus*) spawning during late spring is related to the cumulative sum of degree-days since the previous October when gonad development begins (*Gillet & Quetin, 2006*). Former studies in the Baltic Sea demonstrated a climate triggered regime shift, which resulted in reorganization in the food web as well as warmer winters since the late 1980s (*Mollmann et al., 2008*; *Arula et al., 2014a*; *Arula, Ojaveer &*
*Klais, 2014b*). Warmer winters favored survival of young herring that were recruited to the herring population and increased the relative number of first-time spawners in the population (*Ojaveer et al., 2011*) and accelerated the maturation cycle of young and old herring. Our results show that the differences in K between young and old herring were the greatest after severe winters. Young herring displayed significantly higher K after mild winters compared to severe winters, while winter air temperature did not affect K of repeated spawning older herrings. These results might also explain why the timing of spawning is age-dependent and does not depend on spawning stock size. Our findings, related to fish K after exposure to acute cold stress in severe winters, support conclusions implicating nutritional deficiency as the proximate cause of decreased K, rather than density-dependency before winter since herring spawning stock biomass did not correlate with spawning time or duration. Exposure to winter conditions, both mild and severe, resulted in a decrease in K during the spawning season, most likely attributable to decreased feeding rates at cold temperatures. The mechanism of reduced spawning appears to be low temperatures that are unfavorable for herring feeding (longer fasting periods because herring do not feed at temperatures below 2 degrees C), and there is a reduced rate in accumulation of energy prior to the reproduction and development of its gametes (*Ojaveer, 1988*).

Winter mortality is size-selective, with lasting influences on cohort demographics (*Martino & Houde, 2012*). Small fish are generally more vulnerable because of higher relative energetic demands coupled with a reduced capacity for lipid storage (*Post & Evans, 1989*). On the other hand, in severe winters, an entire cohort is expected to have high mortality rates, regardless of body size (*McCollum, Bunnell & Stein, 2003*; *Michaletz, 2010*). Therefore, it is plausible that density-dependent processes will have a greater impact on overwinter survival during mild or moderate winters. While we did not measure the density-dependency in terms of growth rates and feeding conditions of herring before winter, we observed that young herring had the poorest K after cold winters, implying that despite feeding conditions prior to winter, winter air temperature determines spawner K and has a more pronounced effect for younger individuals. Since young herring in our study denote first-time spawners that began maturation most recently, these results are in contrast to *McCollum, Bunnell & Stein (2003)*, which shows that severe winters will have equal effect for an entire population and across body sizes (as a proxy of age). Age of fish is an important and main determinant, regulating the condition and timing of spawning, and can likely reflect overwintering mortality rates that vary among ages.

It has been demonstrated that the annual maturation process of herring is size-specific rather than age-specific (*Lambert, 1987*; *Ware & Tanasichuk, 1989*) and herring tend to school according to size (see references in *Lambert (1987)*) rather than age. These aggregations are more exaggerated during spawning due to additional behavioral cues associated with the actual spawning act. It is possible that herring of a large year class tend to "entrain" individuals of adjacent year classes. Thus, fish above and below the modal length of the dominant year class would tend to aggregate. This effect is possibly diminished below a certain population density. We show that, despite previous evaluations, Baltic herring spawning shows a certain temporal pattern in terms of consecutive age groups that migrate

to spawn. We found that old, repeat spawners dominate at the beginning of spawning season while young, first time spawning individuals followed older herring cohorts and spawn later in the season. Some studies show that young, first time spawning herrings in the North Sea arrive on spawning grounds before repeat spawners (*Lambert, 1990*). However, we did not observe such a pattern in the GoR herring. It has been suggested that exceptions are caused by different age groups inhabiting separate geographical regions prior to spawning migration (*Lambert, 1987*). Since the distinct Baltic herring populations (GoR and Central Baltic Herring), have different thermal preferences but the same spawning grounds in the NE GoR, mild winter air temperature results in a more even spawning period, but forces distinct populations with different adaptations to utilize the same spawning season after the coldest winters. The potential effect of the population structure of spawning schools of herring on the timing of spawning, structure, and the survival of the annual larval component in the area deserves further investigation.

## CONCLUSIONS

1. The start-date of the herring spawning season varied by months, starting earlier after mild winters, and can be predicted accurately from preceding winter air temperatures.
2. As the spawning season progressed, successively younger individuals replaced older spawners; this pattern varied interannually in response to preceding winter air temperature.
3. Climate-driven regime shifts in the Baltic Sea have resulted in milder winters, but the start-date of the herring spawning season has not changed, indicating non-linear response of Baltic herring spawning to climate warming.

## ACKNOWLEDGEMENTS

The authors are grateful to the editors, two anonymous reviewers and Edward D. Houde for sharing insight in earlier versions of the manuscript.

### Funding

This work was supported by the Baltic American Freedom Foundation (No. P310H9) by providing a personal grant for Timo Arula and by the European Maritime and Fisheries Fund (EMFF) through the European Data Collection Framework and were received during this study. There was no additional external funding received for this study The funders had no role in study design, data collection and analysis, decision to publish, or preparation of the manuscript.

### Grant Disclosures

The following grant information was disclosed by the authors:
Baltic American Freedom Foundation: No. P310H9.
European Maritime and Fisheries Fund (EMFF) through European Data Collection Framework.

## Competing Interests

The authors declare there are no competing interests.

## Author Contributions

- Timo Arula conceived and designed the experiments, performed the experiments, analyzed the data, contributed reagents/materials/analysis tools, prepared figures and/or tables, authored or reviewed drafts of the paper, approved the final draft, drafting manuscript.
- Heli Shpilev performed the experiments, contributed reagents/materials/analysis tools, authored or reviewed drafts of the paper, approved the final draft, data collection.
- Tiit Raid and Elor Sepp performed the experiments, contributed reagents/materials/-analysis tools, authored or reviewed drafts of the paper, approved the final draft.

## Animal Ethics

The following information was supplied relating to ethical approvals (i.e., approving body and any reference numbers):

All the individuals collected and analysed in the present study are conducted as a part of the national data collection framework.

## Data Availability

Raw data file in the Supplemental Materials contains all individuals collected and analysed in present study.

## Supplemental Information

Supplemental information for this article can be found online at http://dx.doi.org/10.7717/peerj.7345#supplemental-information.

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
