# Peer review of "Thermal conditions and age structure determine the spawning regularities and condition of Baltic herring (Clupea harengus membras) in the NE of the Baltic Sea"

_PeerJ, doi:10.7717/peerj.7345_

## Round 0.1 · original submission · Major Revisions

I recommend MAJOR revision, in light of two highly contrasting expert reviews.

My major additional comment is that the language needs to be more accessible; one reviewer expressed concern that the text had simply been copied – please paraphrase. There are too many acronyms used in sentences that makes the narrative difficult to understand – e.g. ‘The BoHS season varied from mid-April (cw 15) in 2015 to mid-May (cw 20) in 2003 (Fig. 2).’ This could be easier to interpret if written as the start of the herring season showed interannual variability, occurring between calendar week 15 and 20 … particularly in light of ‘calendar week’ being spelled out on line 234!

Another example on line 246: ‘Generally, OH had the highest K after cold winters, while in similar conditions YH experienced the poorest K (Fig. 5).’

The English grammar also needs checking.

Line 140-142 ‘The sum was preferred to average values because the cumulative effect does not smooth data over a longer period (see also Ojaveer et al., 2011)’ makes no sense.

Line 110 – maturing instead of pre-spawning. If you’re going to define terms, then please use them.
The authors need to provide a better rationale for how the temporal comparison is possible with ~40 years of data in 1947-1986, compared to ~15 y of data (1999 – 2015). Why were these years analysed: 1947-49, 1953-1955, 1964-66, 1974-76, and 1984-86?

Line 288 – were fish larvae collected in this study? Where is the SSB data presented? The conclusions are not evident in the data presented – needs clarification.

Figure 1 – where is the Gulf of Riga?
Figure 3 – please use the acronyms after the words so that readers can cross-reference to the text
Figure 5 and 6 – define x-axis titles. Include K in y-axis description

Reviewer 1 ·

Basic reporting

- The manuscript includes several violations of plagiarism mostly, e.g. Line: 53-55, Line: 55-57 or Line 89-92
- Refer to own papers, even if they are not the original source
- The reference list is not complete and inconsistent

Experimental design

It is not well described how the herring data was collected. The beginning of the spawning season is defined as the appearance of pre-spawning herring. This is contradictory, either herring are spawning or not. When herring are in pre-spawning conditions, especially on a 6 maturity stage scale, this can be a quite long time. Also using commercial catches is not the optimal solution to define spawning since fishermen only start fishing when sufficient numbers of herring occur. It does not indicate whether herring are caught throughout the whole year. The same account for the larval period, it is not mentioned how they were sampled.

Validity of the findings

Thanks a lot for sharing the raw data. However, the results presented do not correspond with the provided raw data, e.g. total number mentioned in the text 12 148 individuals is not the sum of the individuals in Table 1 (12 031), or mean WAT for week 18 is -13.3 but in Figure 3 it is ~-8.

Additional comments

Given the fact of plagiarism and the fraud of data, I recommend rejecting the manuscript. I like the idea of the manuscript and the results are worth for publishing, however, at the present state, the interpretation of the results is not very convincing.

Reviewer 2 ·

Basic reporting

The paper is well written and structured.

Experimental design

The exp design is sound. Some additional comments on how the environmental proxy project into the ocean, and further how this proxy relates to the larger scale variability would be welcomed

Validity of the findings

Ok

Additional comments

Review of “Thermal conditions and age structure determine the spawning regularities and condition of Baltic herring (Clupea harengus m.) in the NE of the Baltic Sea
Arula et al.

In general, I think this a well written paper. It is easy to follow and seem logical. In think the paper is worth publishing but I have some points that I would like the authors to consider.

General comments
Given my initial background in physical oceanography and ocean climate I am left with the question how much of the observed oceanic warming in the North Atlantic is reflected in the Baltic or GoR. Or is the Baltic climate basically determined by local atmospheric forcing including air-ocean heat fluxes. Also, there should be some discussion about the sensitive of proxy to the choice period (herein Jan-Mar)?

The paper show that the integrated Tair is a useful proxy for winter severity, however some idea on how this proxy projects into the ocean in terms of temperature variability between cold/warm years would be relevant.

Details

Introduction
Line 44-45 The sentence “Still, not all fish stocks are so well predictable with high accuracy”. This sentence seems a bit strange, basically most fish species are not well predictable. Also it somewhat unclear, i.e. predictable in terms of what? Please clarify.

Line 85-86: need to specify what region what is referred to here. In the air or ocean, and preferable also how much.

Line 104: define calendar week (CW) here?

Line 189-190: does the factor 100 have any effect on the statical results, is this needed?

236-237: for readability, you correlate mean average age of herring with calendar week of what? Please include.

247-248: I do not understand what is mean here, can you clarify, add some explanation?

273-274: Do this imply that also suitable prey is delayed during a cold winter, is it observed.

277-279: Does this mean that the Baltic show large inter-annual variability but no trend in WAT temperature changes? Clarify

278-279: which two periods , please specify

298-299: maybe change “ BoHS in later season” to “ BoHS later in the season”, please check

354-355: Missing word “ …it results in a more .. “

---

## Round 0.2 · Minor Revisions

I am sorry for the delay, but in addition to difficulties in locating suitable referees, the Academic Editor handling your submission resigned. To expedite your decision, I have reviewed the previous versions, and the reviews to date. I believe your manuscript will be acceptable for publication following minor revisions. In addition to the feedback from the reviewer, I have attached an annotated PDF with some suggested revisions, none of which seem particularly difficult, so I do not expect that you should have any trouble in turning this around quickly. Please feel free to contact me if you have any questions about my recommendations or need any assistance in completing the revisions to your manuscript.

Reviewer 2 ·

Basic reporting

Line 108-109: Please rephrase " ... twice warmer .. " which does not make sense.

Make sure refs are correct, e.g.
line 335: (Arula 2014 should be Arula et al., ..

Conclusions points 1 and 3 should be written more clearly.

Experimental design

No comment

Validity of the findings

No comment

Additional comments

Thanks for clarifying my points.
Also check the text for typos.

---

## Round 0.3 · accepted · Accept

Thank you for making the final requested edits and clarifications to your submission. Given the suggested changes and clarifications to the text have been incorporated to the revised manuscript, and the raw data are now referenced in the paper and included in the appendices, I see no reason to delay your acceptance any longer. Congratulations, and I look forward to seeing your manuscript come out in press.

#